# Structural mechanisms of the human cardiac sodium-calcium exchanger NCX1

Jing Xue [1,2], Weizhong Zeng[1,2], Yan Han [1,2], Scott John [3], Michela Ottolia[4] & Youxing Jiang [1,2] ✉

Na$^+$/Ca$^{2+}$ exchangers (NCX) transport Ca$^{2+}$ in or out of cells in exchange for Na$^+$. They are ubiquitously expressed and play an essential role in maintaining cytosolic Ca$^{2+}$ homeostasis. Although extensively studied, little is known about the global structural arrangement of eukaryotic NCXs and the structural mechanisms underlying their regulation by various cellular cues including cytosolic Na$^+$ and Ca$^{2+}$. Here we present the cryo-EM structures of human cardiac NCX1 in both inactivated and activated states, elucidating key structural elements important for NCX ion exchange function and its modulation by cytosolic Ca$^{2+}$ and Na$^+$. We demonstrate that the interactions between the ion-transporting transmembrane (TM) domain and the cytosolic regulatory domain define the activity of NCX. In the inward-facing state with low cytosolic [Ca$^{2+}$], a TM-associated four-stranded β-hub mediates a tight packing between the TM and cytosolic domains, resulting in the formation of a stable inactivation assembly that blocks the TM movement required for ion exchange function. Ca$^{2+}$ binding to the cytosolic second Ca$^{2+}$-binding domain (CBD2) disrupts this inactivation assembly which releases its constraint on the TM domain, yielding an active exchanger. Thus, the current NCX1 structures provide an essential framework for the mechanistic understanding of the ion transport and cellular regulation of NCX family proteins.

Sodium-calcium exchangers (NCX) are membrane antiporters that control the extrusion or entry of Ca$^{2+}$ across the cell membrane in exchange for Na$^+$[1-4]. They are ubiquitously expressed and play a central role in maintaining cellular calcium homeostasis for cell signaling[5,6]. Dysfunctions of NCXs are associated with many human pathologies, including cardiac hypertrophy, arrhythmia, and postischemic brain damage[1,7-9]. NCX-catalyzed ion exchange reaction is electrogenic with a stoichiometry of 3 Na$^+$ for 1 Ca$^{2+}$. NCX normally functions to extrude Ca$^{2+}$, but it can be reversed depending on the chemical gradient of Na$^+$ and Ca$^{2+}$ and the membrane potential[1,10-16]. Three NCX isoforms (NCX1–3) are found in mammals, and each isoform carries various splice variants that are expressed in specific tissues and exhibit distinct regulatory phenotypes[2,17-22]. The cardiac exchanger NCX1.1 has been the most extensively studied variant whose function is central to the cardiac contractile activity[23-26].

The eukaryotic NCX consists of a TM domain with 10 TM helices and a large intracellular regulatory domain which, in primary sequence, separates the TM domain into two homologous halves (TMs 1–5 and TMs 6–10)[2,27-30]. The TM domain is responsible for the ion exchange function in NCX[29,31] and has been modeled by the crystal structures of the archaebacterial exchanger NCX_Mj[30,32]. The intracellular domain is involved in the allosteric regulation of the exchanger by cytosolic Ca$^{2+}$ and Na$^+$[29,31]. In cardiac NCX1, cytosolic Ca$^{2+}$ binds to two calcium-binding domains (CBDs) within the intracellular domain to

[1]Howard Hughes Medical Institute and Department of Physiology, The University of Texas Southwestern Medical Center, Dallas, TX, USA. [2]Department of Biophysics, The University of Texas Southwestern Medical Center, Dallas, TX, USA. [3]Department of Medicine (Cardiology), David Geffen School of Medicine, University of California Los Angeles, Los Angeles, CA, USA. [4]Department of Anesthesiology and Perioperative Medicine, Division of Molecular Medicine, David Geffen School of Medicine, University of California Los Angeles, Los Angeles, CA, USA. ✉e-mail: youxing.jiang@utsouthwestern.edu

increase transport activity[33–35], while cytosolic Na⁺ inhibits transport activity by driving the exchanger into an inactivated state known as Na⁺-dependent inactivation[16,31,36,37]. The Na⁺-dependent inactivation occurs when three Na⁺ ions are bound to the inward-facing exchanger[37–39]. An amphipathic region, known as eXchanger Inhibitory Peptide (XIP) at the N-terminus of the intracellular domain, plays a central role in the inactivation process[40,41]. NCX1 activity can also be modulated by PIP2[36,42–44] as well as posttranslational modification such as palmitoylation[45–48].

Despite the extensive investigations, little is known about the architecture of the eukaryotic NCX1 and the coupling between its TM and intracellular regulatory domains. Here we present the structures of human cardiac NCX1 in both the inactivated and Ca²⁺-activated states, revealing the molecular basis underlying the complex regulation of NCX1.

## Results

### Structure determination of human cardiac NCX1

Human cardiac NCX1 was expressed in HEK293S cells using the Bac-Mam system and purified in digitonin detergent. Preliminary cryo-EM data for the full-length exchanger did not yield useful 3D reconstructions, likely because of the small size of the exchanger protein along with a highly mobile cytosolic domain. We reasoned that shortening the flexible loop between the TM and cytosolic domain could help stabilize the exchanger. Therefore, we generated a series of deletion constructs targeting the predicted internal loop between XIP and CBD1. One deletion mutant, Δ341-365, significantly improved the particle alignment, yielding a 3D map at 4.1 Å. For further improvement, we raised a monoclonal antibody against NCX1 and used its Fab fragment as a fiducial marker to facilitate the single particle alignment ("Methods"). Using the protein sample prepared in 200 mM Na⁺ and nominal Ca²⁺-free condition, the structure of NCX1 in complex with Fab was determined to an overall resolution of 3.1 Å ("Methods", Supplementary Figs. 1–3 and Supplementary Table 1). Focused refinement of the TM and cytosolic domains further improved the quality of the EM density maps for the respective regions, allowing for accurate model building for the majority of the protein (Fig. 1a, Supplementary Movie 1 and Supplementary Fig. 4). To ensure that the deletion mutant used for structure determination does not compromise the exchanger function, we utilized the giant patch technique to characterize its biophysical properties[35,49,50] and demonstrated that the deletion mutant is fully functional and exhibits a similar response to both cytosolic Ca²⁺ and Na⁺ as the wild-type exchanger (Fig. 1c).

As expected, the human NCX1 contains two major structural components, the TM domain for ion exchange and the cytosolic regulatory domain (Fig. 1a, b). The TM domain shares an analogous 10-TM topology as NCX_Mj and can be divided into two 5-TM halves with similar structures but inverted topology. The two CBDs constitute the major part of the cytosolic domain and adopt an elongated, shallow V-shaped conformation with a hinge angle of 120°. CBD1 is positioned distal to the TM domain, and its N-terminus extends far into the cytosol and is connected to the XIP region by a long disordered loop. CBD2 is proximal to the TM domain, and its C-terminal region makes direct contact with the TM domain and becomes a central part of inter-domain contact responsible for NCX inactivation. The antibody Fab fragment binds at the N-terminal half of CBD2. For simplicity, we only modeled the Fv part of the Fab fragment as its density is well-defined in the EM map. As discussed in detail later, the overall structure of NCX1 represents an inward-facing, inactivated state.

### The inward-facing TM domain of NCX1

The TM domain of NCX1 shares a similar overall architecture to the archaea exchanger NCX_Mj, suggesting the structural conservation of the TM module for ion transport function among all NCXs[30]. Eight of the TM helices (TMs 2–5 and 7–10) form a tightly packed core with two highly conserved α-repeats (TMs 2–3 as α1 and TMs 7–8 as α2) bundled at the center; TMs 1 and 6 are much longer helices at the periphery of the TM domain and loosely pack against the core at a 45° angle (Fig. 2a). Conserved residues from both α-repeats constitute the central ion binding sites almost identical to those in NCX_Mj, which were defined as $S_{ext}$ and $S_{int}$ for Na⁺ binding, $S_{Ca}$ for Ca²⁺ or Na⁺, and $S_{mid}$ for a water molecule (Fig. 2b)[30,32]. Contrary to the outward-facing NCX_Mj whose external $S_{ext}$ site is connected to the extracellular side via a solvent-accessible passageway, NCX1 adopts an inward-facing conformation with its internal $S_{int}$ site connected to a passageway for intracellular ion access while its $S_{ext}$ is completely occluded from the external side (Fig. 2c and Supplementary Fig. 5a). Prepared in 200 mM Na⁺ with nominally free Ca²⁺, most of the exchangers are likely in a Na⁺-bound state, despite only weak density at $S_{Ca}$ is visible within the central ion binding sites (Fig. 2b). Therefore, three Na⁺ ions and a water molecule are modeled in NCX1 based on the well-defined ion-ligand geometry in the NCX_Mj structures[30,32]. It is worth noting that an inward-facing NCX1 is known to have a high Ca²⁺ affinity for its ion-exchanging site[16,32,51,52], and micromolar range of Ca²⁺ is still present in the nominally Ca²⁺-free solution[53]; we suspect that the $S_{Ca}$ sites in a small portion of NCX1 proteins are Ca²⁺ occupied, which may explain the relatively stronger density at $S_{Ca}$.

Other than the peripheral TMs 1 and 6, the core part of the NCX1 TM domain (TMs 2–5 and TMs 7–10) superimposes well with that of outward-facing NCX_Mj (Fig. 2d), suggesting that the core remains largely static during ion exchange whereas TMs 1 and 6 undergo a major conformational change. The different positions of TMs 1 and 6 between the two structures imply a simple sliding motion of these two helices upon inward-outward conformational transition. This is consistent with the previous structural analysis of NCX_Mj, in which an inward-facing model was predicted based on the structural homology between the two halves of the TM domain, and a sliding-door motion of TMs 1 and 6 during ion exchange was proposed for rapid alternative access to the ion transport sites[30]. Indeed, the NCX1 TM domain is strikingly similar to the predicted inward-facing model of NCX_Mj (Supplementary Fig. 5b).

### Key auxiliary components of the TM domain

Several key structural elements are directly associated with the TM domain and are important for NCX1 regulation (Fig. 2e). The first is a 4-stranded β-sheet structure, termed β-hub, right underneath the TM domain. Two of the strands (β1 and β2) come from the cytoplasmic loop between TMs 1 and 2 and form a curved β-hairpin extending into the cytosol. The other two strands (β3 and β4) are formed by the post TM5 region from amino acids 219–248, named XIP region as it includes the classically defined XIP sequence (a.a. 219–238). Rather than an unstructured loop as initially predicted, the major part of the XIP region forms a twisted antiparallel β-sheet that runs parallel to the internal membrane surface. Through parallel β-sheet packing, the β-hairpin and the XIP region assemble into a C-shaped clamp that serves as the hub for inter-domain interactions between the TM and cytosolic domains in inactivated NCX1 as described later. The second element consists of the fifty amino acids right before TM6 (a.a. 708–757) initially predicted to be part of the cytosolic loop domain[54]; it forms two arch-shaped helices (TMH1 and TMH2) partially embedded in the inner leaflet of the membrane (Fig. 2e, right inset). The C-terminal half of TMH2 makes extensive hydrophobic contact with the cytosolic portion of TMs 1 and 6, suggesting that TMH1 and TMH2 are an integral part of the TM domain and likely move together with TMs 1 and 6 during inward–outward conformational changes. C738 at the N-terminus of TMH2 is the palmitoylation site important for NCX1 trafficking and regulation[45,46,48]. Thirdly, we observed a tightly bound Ca²⁺ ion at the short extracellular loop between TMs 4 and 5 (Fig. 2e, left inset), which, interestingly, was also observed at the equivalent site in the structure of NCX_Mj[30]. While it is unclear whether this external

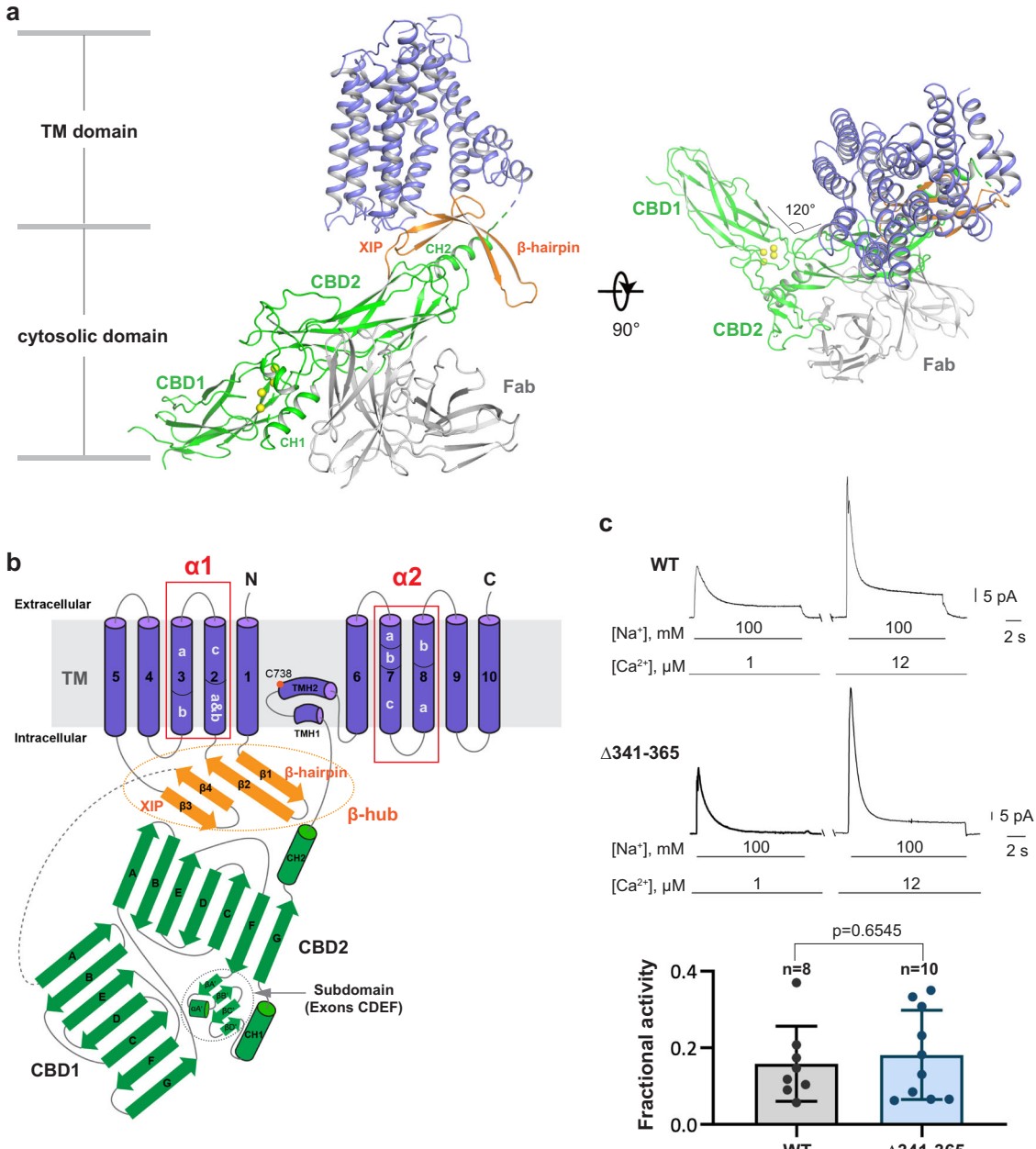

**Fig. 1 | Overall structure of human cardiac NCX1. a** Cartoon representation of the NCX1 structure obtained in high Na$^+$ and low Ca$^{2+}$ conditions with the TM domain colored blue, XIP and β-hairpin colored orange, and CBD colored green. For the Fab antibody, only the Fv fragment (gray) is modeled. Yellow spheres represent the bound Ca$^{2+}$ in CBD1. **b** Topology diagram of NCX1. All α-repeat helices are bent into two or three segments marked as a, b, or c on TMs 2, 3, 7, and 8. **c** Representative outward currents recorded from oocytes expressing the WT and Δ341–365 NCX1 constructs. Lines below the traces indicate solution changes with the indicated concentrations. Fractional activity was measured as the ratio between steady state and peak currents recorded in the presence of 12 μM cytosolic Ca$^{2+}$. $n$ = independent experiments. Errors represent mean ± SEM. $p = 0.6545$ by a two-sided unpaired $t$-test. All recordings were performed at 35 °C with a holding potential of 0 mV. Source data are provided as a Source Data file.

Ca$^{2+}$ site plays any physiological role, mutations at Ca$^{2+}$-chelating E199 significantly reduce the NCX1 activity[55], implying that external Ca$^{2+}$ binding could be important for NCX1 function.

### The cytosolic domain of NCX1

The two homologous Ca$^{2+}$ binding domains (CBD1 and CBD2) constitute the major part of the cytosolic domain (Fig. 3a). Each CBD has a classical immunoglobulin fold formed by two layers of antiparallel β-sheets consisting of strands A-B-E and strands D-C-F-G, respectively[34,56,57]. Despite nominal Ca$^{2+}$-free conditions, the distal CBD1 is in a Ca$^{2+}$-bound state with all four of its Ca$^{2+}$-binding sites

occupied, indicating high Ca$^{2+}$ affinity (Fig. 3a, left inset). Clustered near the interface between CBD1 and CBD2, the bound Ca$^{2+}$ ions stabilize the local structure, which in turn facilitates the inter-domain interactions between the CBDs. With its four sites distant from the TM domain, CBD1 Ca$^{2+}$ binding may not directly impact the ion exchange function of the TM domain. One possibility is that Ca$^{2+}$ binding at CBD1 mainly plays a structural role and rigidifies the hinge between the two CBDs, as previously demonstrated in the study of isolated CBD1–CBD2 domain[58–61].

The proximal CBD2 domain is more complex and contains some extra structural elements in addition to the immunoglobulin fold.

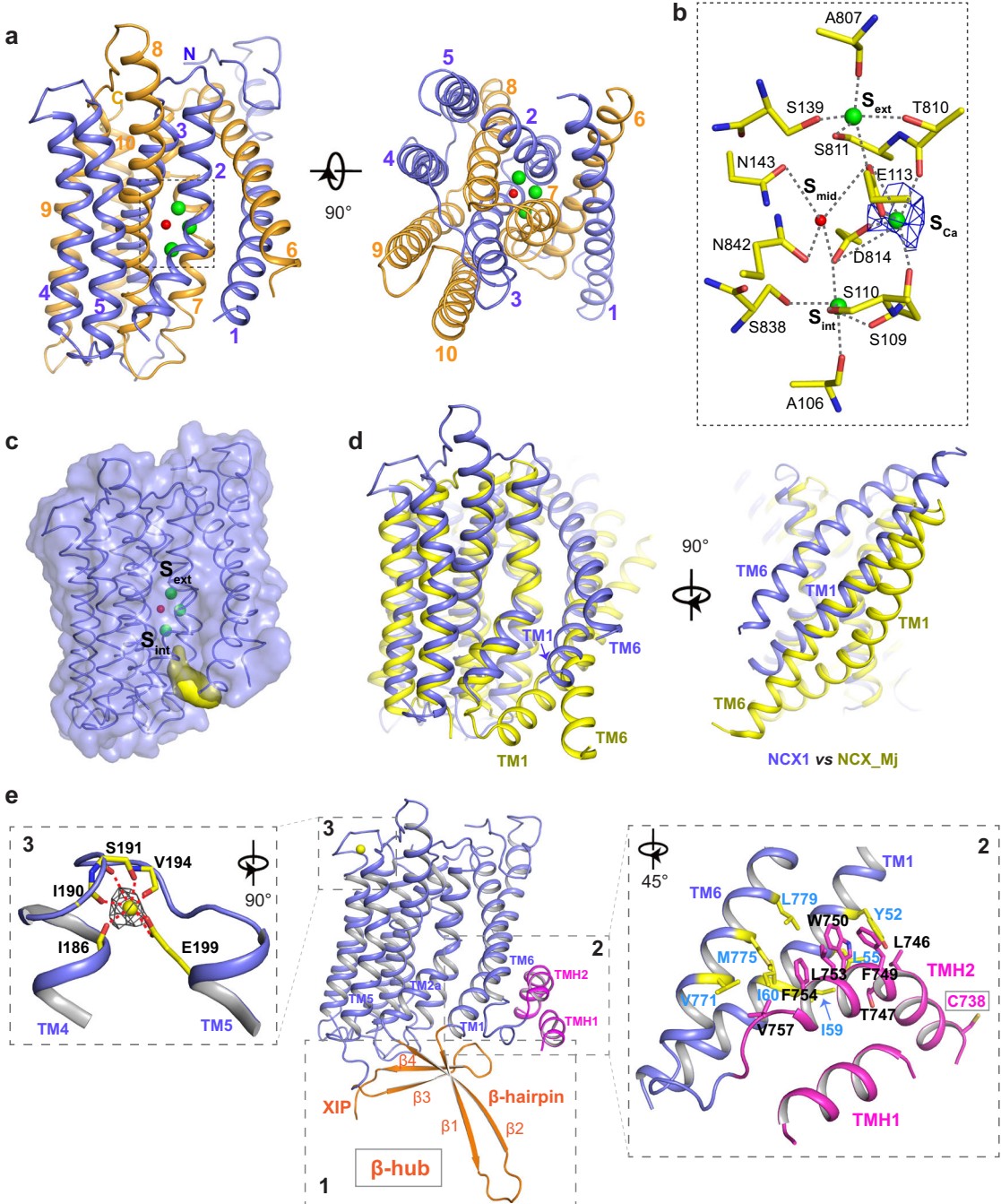

**Fig. 2 | Structure of NCX1 TM domain. a** Side and bottom views of the NCX1 TM domain with three Na⁺ ions (green spheres) and a water molecule (red sphere) modeled at the central sites (boxed region). The N- and C-terminal halves are colored blue and orange, respectively. **b** Molecular details of the ion binding sites with EM density (blue mesh) contoured at $7\sigma$. **c** Surface rendered NCX1 TM domain showing the internal passage (yellow) to $S_{int}$ site. **d** Structural comparison between the NCX1 TM domain (blue) and the outward-facing NCX_Mj (yellow, PDB: 3V5U). **e** Three key auxiliary components of the TM domain. Insets are zoomed-in views at the TMH1/2 region and the external $Ca^{2+}$ (yellow sphere) site with its density (gray mesh) contoured at $20\sigma$.

Residues 597–644 right after βF form a subdomain consisting of a short helix and four short β-strands. The subdomain is positioned near the inter-CBD hinge, fastening the two CBDs by directly interacting with both (Fig. 3a, right inset). It is worth noting that the NCX1 gene (SLC8A1) contains a cluster of exons (A–F) whose combination generates various splice variants. NCX1.1 is the longest variant containing exons A and CDEF[4,19,62]. The subdomain is encoded by exons CDEF and represents an NCX1.1-specific structure feature. CBD2 also contains two helices, termed CH1 and CH2, right before and after βG, respectively. The CH2 helix is partially cuffed by the C-shaped β-hub and becomes a central part of the inter-domain interactions that stabilize

NCX1 in the inactivated state (Fig. 3b). The two CBD2 $Ca^{2+}$ sites are in the apo state, indicating lower affinity than those in CBD1. Near the interface between CBD2 and the β-hub, $Ca^{2+}$-induced conformation changes in CBD2 destabilize the inactivation assembly, as discussed below.

## Inactivation assembly of NCX1

The TM and cytosolic domains of eukaryotic NCX were initially predicted to be two loosely attached entities[63,64]. The current structure of NCX1, however, demonstrates that these two parts form a tightly packed structural unit that stabilizes the exchanger in the inactivated

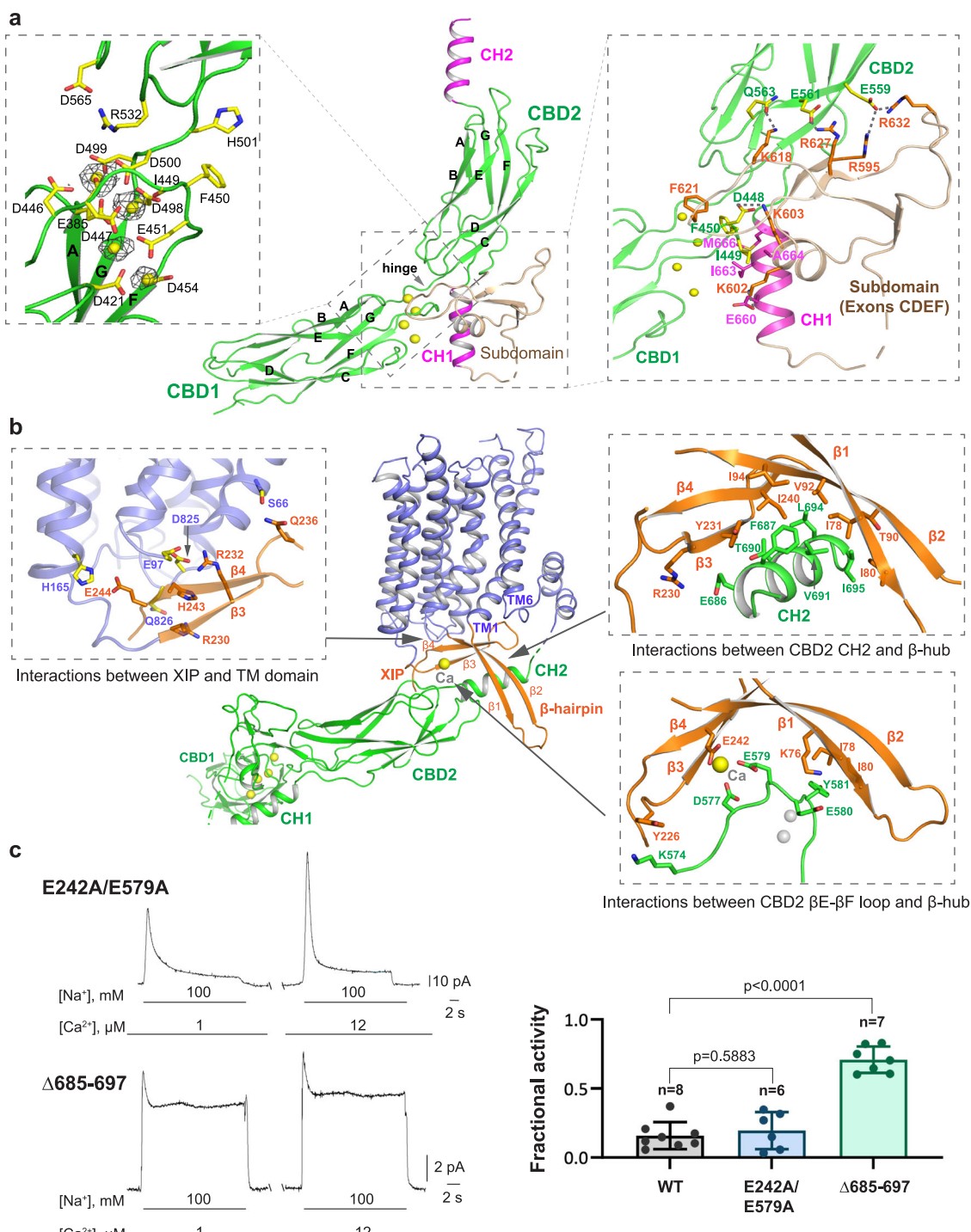

**Fig. 3 | The cytosolic domain and the inactivation assembly. a** Overall structure of the cytosolic domain with zoomed-in views of the CBD1 Ca²⁺ sites and CBD2 subdomain. The density of the bound Ca²⁺ ions (yellow spheres) is shown as gray mesh (contoured at 21σ). The CBD2 subdomain (light brown) is encoded by exons CDEF. **b** Molecular details of the β-hub-mediated inter-domain interactions in the inactivation assembly. The locations of two CBD2 Ca²⁺ sites are marked by the two gray spheres in the lower right inset. **c** Representative outward currents

recorded from oocytes expressing the E242A/E579A and Δ685−697 mutant constructs and their fractional activities at 12 μM cytosolic Ca²⁺. $n$ = independent experiments. Errors represent mean ± SEM. P values were calculated by a two-sided unpaired t-test (p = 0.5883 for WT vs. Δ685−697 and $p < 0.0001$ for WT vs. E242A/E579A). All recordings were performed at 35 °C with a holding potential of 0 mV. Source data are provided as a Source Data file.

state (Fig. 3b). The interactions between these two domains are mediated by the apo-state CBD2 and the C-shaped β-hub formed by the β-hairpin (β1 and β2) and the XIP β-sheet (β3 and β4). Within the β-hub, the XIP β-sheet runs parallel to the membrane and anchors the hub to the TM domain through mostly hydrophilic interactions (H-bonds and salt bridges) (Fig. 3b, left inset). As the β-hairpin is directly

linked to TM1 and expected to move along with it during inward-outward conformational changes, the β-hub can only assemble when the TM domain in an inward-facing conformation, otherwise the β-hairpin and TM1 would collide directly with XIP when moving towards the outward-facing conformation (Supplementary Fig. 5c, d and Supplementary Movie 2). The prerequisite of inward-facing NCX for β-hub

formation and the resulting inactivation assembly with CBD2 is consistent with the prediction that cytosolic $Na^+$-dependent inactivation occurs when the exchanger is in the inward-facing conformation and likely in a $Na^+$-loaded state[37–39].

The CH2 helix from CBD2 is partially encircled by the β-hub and contributes to the majority of inter-domain contact by engaging in hydrophobic and hydrophilic interactions with both the β-hairpin and XIP (Fig. 3b, upper right inset). The loop between βE and βF that encompasses the two CBD2 $Ca^{2+}$ sites also participates in the inter-domain interactions with the β-hub (Fig. 3b, lower right inset). Notably, part of these interactions is mediated by a cation that bridges three acidic residues (D577, E579, and E242). The strategic location of this cation-mediated interaction between CBD2 and XIP led to our earlier speculation that this could be a $Na^+$ site important for $Na^+$-dependent inactivation. However, two observations negate our hypothesis: chelated by three acidic residues, the chemical environment of this bound cation is optimal for $Ca^{2+}$ rather than $Na^+$; the E242A/E579A double mutant exhibits similar $Na^+$-dependent inactivation as the wild-type exchanger in electrophysiological recordings (Fig. 3c). Whether this inter-domain $Ca^{2+}$ site plays any functional role in NCX warrants further study.

Thus, the extensive interactions between the apo CBD2 and the β-hub define the stable inactivation assembly that locks TMs 1 and 6 in the inward-facing conformation and stalls the ion exchange activity of NCX1. Mutations that change the stability of the inactivation assembly are expected to have a profound effect on NCX activity. Most mutagenesis studies on NCX1 $Na^+$-dependent inactivation have been centered on the XIP region, yielding either gain or loss of function phenotypes[40]. Those mutations that stabilize or destabilize the inactivation assembly likely promote or mitigate the inactivation, respectively. The NCX1 structure would also predict an essential role of the CH2 helix in forming a stable inactivation assembly. Indeed, the removal of CH2 yields a highly active exchanger that loses most of its cytosolic $Ca^{2+}$ activation and $Na^+$-dependent inactivation properties (Fig. 3c), similar to chymotrypsin-treated NCX1, whose cytosolic regulatory module is enzymatically removed[31].

### Structural mechanism of cytosolic $Ca^{2+}$ activation

A comparison between the apo CBD2 in our NCX1 structure and the $Ca^{2+}$-bound CBD2 crystal structure (PDB: 2QVM)[57] suggests the potential $Ca^{2+}$-induced conformational changes activate NCX1 by destabilizing the inactivation assembly (Fig. 4a and Supplementary Movie 3). The main structural changes occur at the $Ca^{2+}$-binding loop (a.a. 578–584) between βE and βF as well as the C-terminus of βG, both of which undergo a concurrent swing motion from apo to $Ca^{2+}$-bound state. This swing motion is not a simple rigid body movement but involves main-chain and side-chain rearrangement at defined residues, which can be summarized into three key movements. First, the side chains and peptide bonds of both E582 and K583 flip between apo and $Ca^{2+}$-bound states. K583 sidechain, initially pointing towards $Ca^{2+}$ site 2 and making direct interaction with D552 in the apo state, subsequently flips away from D552 towards the bulk solution upon $Ca^{2+}$ binding. Conversely, the E582 sidechain, which is initially exposed to the bulk solution in the apo state, subsequently flips inwardly to form a salt bridge with K585 upon $Ca^{2+}$ binding. The side-chain rearrangement of both residues liberates D552 from salt-bridging with K583 and K585, allowing it to directly chelate site 2 $Ca^{2+}$. Second, the swing motion of E580 and its two neighbors (E579 and Y581) from apo to $Ca^{2+}$-bound state removes its occlusion of site 1 $Ca^{2+}$. Third, the swing motion of the $Ca^{2+}$ binding loop from apo to $Ca^{2+}$-bound conformation also yields space for the concurrent movement at the C-terminus of βG (a.a. 683-685), allowing the E683 side chain to be re-orientated for direct chelation with site 1 $Ca^{2+}$. Although not present in the CBD2 crystal structure, the CH2 helix immediately following βG is expected to undergo a swing motion similar to the tail of βG (Fig. 4b).

Occurring at the center of the inactivation assembly, the local $Ca^{2+}$-induced CBD2 conformational changes can have a dramatic impact on the stability of this assembly. In the context of the inactivated NCX1 structure, the $Ca^{2+}$-induced local movements would cause a direct collision between CH2 and β-hairpin (Fig. 4b) and likely result in the destabilization of the β-hub as well as the entire assembly. To provide structural insight into the $Ca^{2+}$ activation of NCX1, we tried to obtain its EM structure in the presence of high $Ca^{2+}$. As expected, the exchanger becomes highly dynamic in the presence of $Ca^{2+}$. Nevertheless, we were able to classify the protein particles into two groups and determined their structures (Supplementary Fig. 6). Albeit at a relatively lower overall resolution (3.8 Å), the structure of group I particles is the same as the inactivated NCX1 with its CBD2 tightly attached to the TM domain in a $Ca^{2+}$-free conformation (Fig. 4c and Supplementary Fig. 7a). It is intriguing to observe a portion of the exchangers still in the apo, inactivated state even with the presence of high $Ca^{2+}$. It is possible that both the binding of Fab to CBD2 and the removal of the exchanger protein from its lipid environment in the biochemical purification contribute to the reduction of the CBD2 $Ca^{2+}$ sensitivity. The structure of group II particles represents the $Ca^{2+}$-activated NCX1, whose TM and cytosolic domains are separated from one another and become highly mobile. With the fiducial Fab marker, we were able to obtain a 3.2 Å map for the cytosolic domain using focused classification which reveals a $Ca^{2+}$-bound CBD2 structure (Fig. 4c and Supplementary Fig. 7b). Although the map for the TM region is poorly resolved (~7.6 Å), it contains clear cylinder density for some TM helices, allowing us to define the approximate position of the TM domain (Supplementary Fig. 7c). In comparison with the inactivated state, the group II NCX1 structure illustrates that the entire cytosolic domain of NCX1 undergoes a quite dramatic rigid body movement upon $Ca^{2+}$ activation: it swings counterclockwise, drifts laterally to the edge of the TM domain near the TMH1/2 helices, and undergoes about 130° rotation along a vertical axis (Fig. 4d and Supplementary Movie 4). Thus, the two NCX1 structures likely simulate the steady state NCX1 after $Ca^{2+}$ activation, which is an equilibrium between inactivated and active states. $Ca^{2+}$ binding at CBD2 disassembles the inactivation apparatus and activates the exchanger, whereas $Ca^{2+}$ unbinding at CBD2 allows for the reassembly of the inactivation apparatus and inhibits the exchanger activity.

## Discussion

In summary, we determined the structures of the human cardiac NCX1, which revealed the structural basis underlying $Na^+$-dependent inactivation and cytosolic $Ca^{2+}$ activation in NCX1. Based on the current structures and a plethora of previous electrophysiology data, we propose a working model summarizing the process of $Na^+$-dependent inactivation and the $Ca^{2+}$-induced relief of the inactivation (Fig. 5). NCX1 inactivation commonly occurs when the ion exchange reaction is in the reverse mode ($Na^+$ efflux) with low cytosolic $[Ca^{2+}]$ and the exchanger is in a $Na^+$-loaded, inward-facing state. Only when in this state the linker β-hairpin between TMs 1 and 2ab would be in a position to interact with XIP and form the β-hub. Once formed, the β-hub can readily interact with the cytosolic domain mainly through the CH2 helix of CBD2 and generate a stable inactivation assembly that locks the TMs 1 and 6 in the inward-facing conformation, preventing the TM module from transporting ions. At higher cytosolic $[Ca^{2+}]$, the conformational change induced by $Ca^{2+}$ binding at CBD2 causes a direct collision between its CH2 helix and the β-hub, resulting in the disassembly of the β-hub and the detachment of the entire cytosolic domain from the TM module. This detachment likely also pulls the XIP region away from the TM and releases TMs 1 and 6 from any constraint, yielding a constitutively active TM module for ion transport.

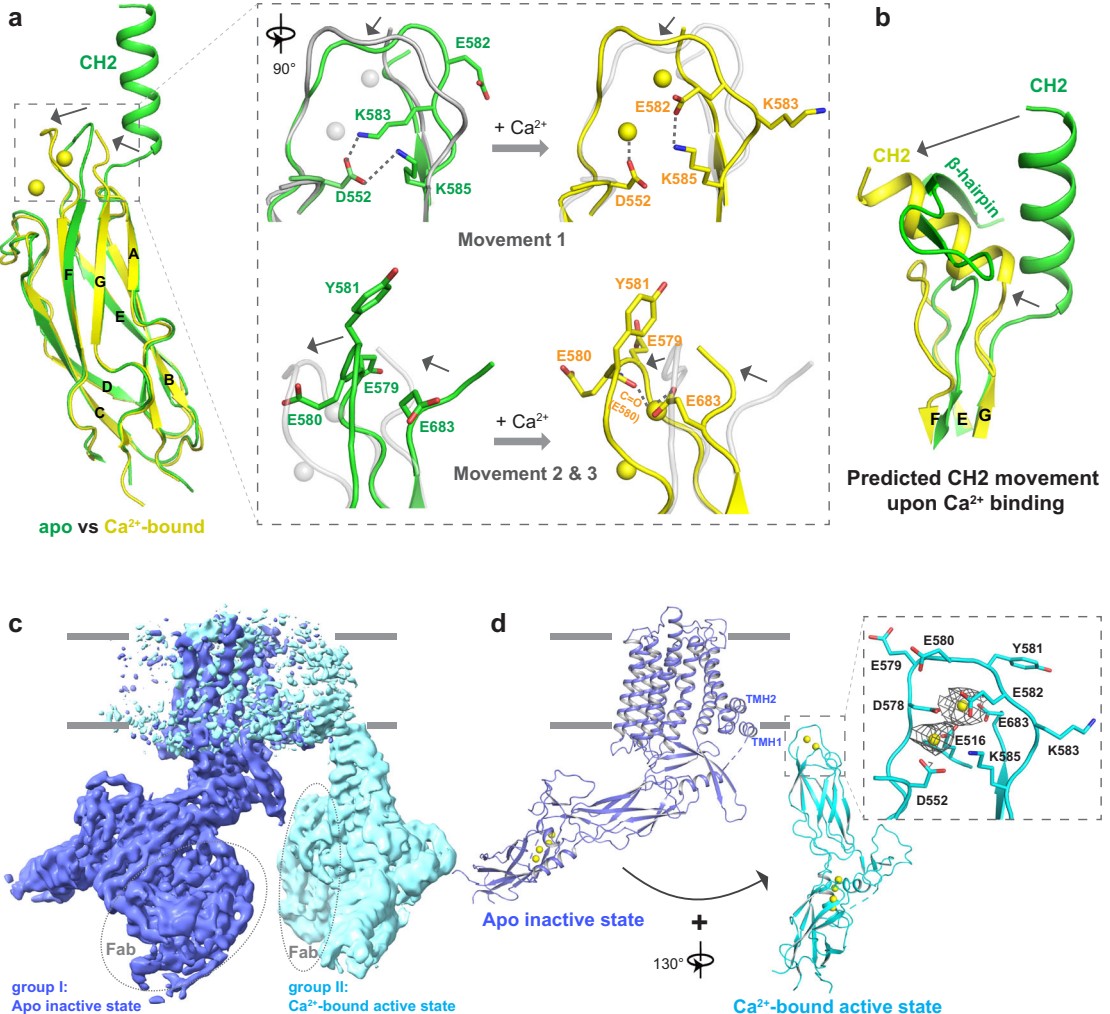

**Fig. 4 | Ca²⁺ induced conformational change in CBD2 and Ca²⁺ activation mechanism. a** Structural comparison between the apo CBD2 from the current NCX1 structure (green) and the crystal structure of the Ca²⁺-bound CBD2 domain (yellow, PDB: 2QVM) with zoomed-in views of the key structural changes from apo (green) to Ca²⁺-bound (yellow) state. Arrows mark the directions of the movements. **b** Predicted CH2 movement along with the C-terminus of βG upon Ca²⁺ binding from apo (green) to Ca²⁺-bound (yellow) state. This movement would result in a direct collision with β-hairpin. **c** EM maps of the two NCX1 structures obtained from the protein sample prepared in the presence of 0.5 mM Ca²⁺. For comparison, the two maps are superimposed at the TM regions. **d** Cartoon representation of NCX1 conformational changes from Na⁺-dependent inactivation state (with apo CBD2, blue) to cytosolic Ca²⁺-activated state (with Ca²⁺-bound CBD2, cyan). The arrow and the rotational angle mark the rigid body movement of the cytosolic domain between the two states. Inset is the zoomed-in view of the CBD2 Ca²⁺ sites illustrating the density (gray mesh contoured at 11σ) from the bound Ca²⁺.

## Methods
### Protein expression and purification
Full-length or truncated human NCX1 (NCBI accession: NP_066920.1) containing a C-terminal Strep-tag was cloned into a pEZT-BM vector[65] and expressed in HEK293S GnTI- cells using the BacMam system (ATCC). Bacmids were synthesized using *E. coli* DH10Bac cells (Thermo Fisher Scientific), and baculoviruses were produced in Sf9 cells using Cellfectin II reagent (Thermo Fisher Scientific). For protein expression, cultured GnTI- cells were infected with the baculoviruses at a ratio of 1:20 (virus: GnTI-, v/v) for 10 h. 10 mM sodium butyrate was then introduced to boost protein expression level, and cells were cultured in suspension at 30 °C for another 60 h and harvested by centrifugation at 4000g for 15 min. All purification procedures were carried out at 4 °C. The cell pellet was resuspended in lysis buffer (25 mM HEPES pH 7.4, 300 mM NaCl, 2 μg/ml DNase I, 0.5 μg/ml pepstatin, 2 μg/ml leupeptin, 1 μg/ml aprotinin, and 0.1 mM PMSF) and homogenized by sonication. NCX1 was extracted with 2% (w/v) N-dodecyl-β-D-maltopyranoside (DDM, Anatrace) supplemented with 0.2% (w/v) cholesteryl hemisuccinate (CHS, Sigma Aldrich) by gentle agitation for 2 h. After extraction, the supernatant was collected by centrifugation at

40,000g for 30 min and incubated with Strep-Tactin affinity resin (IBA) for 1 h. The resin was then collected on a disposable gravity column (Bio-Rad) and washed with 30 column volumes of buffer A (25 mM HEPES pH 7.4, 200 mM NaCl) supplemented with 0.06% (w/v) digitonin (Fisher Scientific) or 0.03% (w/v) lauryl maltose neopentyl glycol (LMNG, Anatrace). NCX1 was eluted in a wash buffer supplemented with 50 mM biotin. The protein eluate was concentrated and further purified by size-exclusion chromatography on a Superdex200 10/300 GL column (GE Healthcare) in buffer A with 0.06% digitonin or 0.03% LMNG.

### Antibody generation and purification of NCX1–Fab complexes
Monoclonal antibodies against full-length human NCX1 were raised by the Vector and Gene Therapy Institute (VGTI) at Oregon Health & Science University (OHSU) using standard methods. The purified NCX1 in LMNG was used for immunization. Antibodies were screened by ELISA using both the native and denatured exchanger proteins as substrate and western blot to select clones that recognized natively folded NCX1. 2E4 IgG was identified as an NCX1-specific monoclonal antibody from the screening, and subsequent

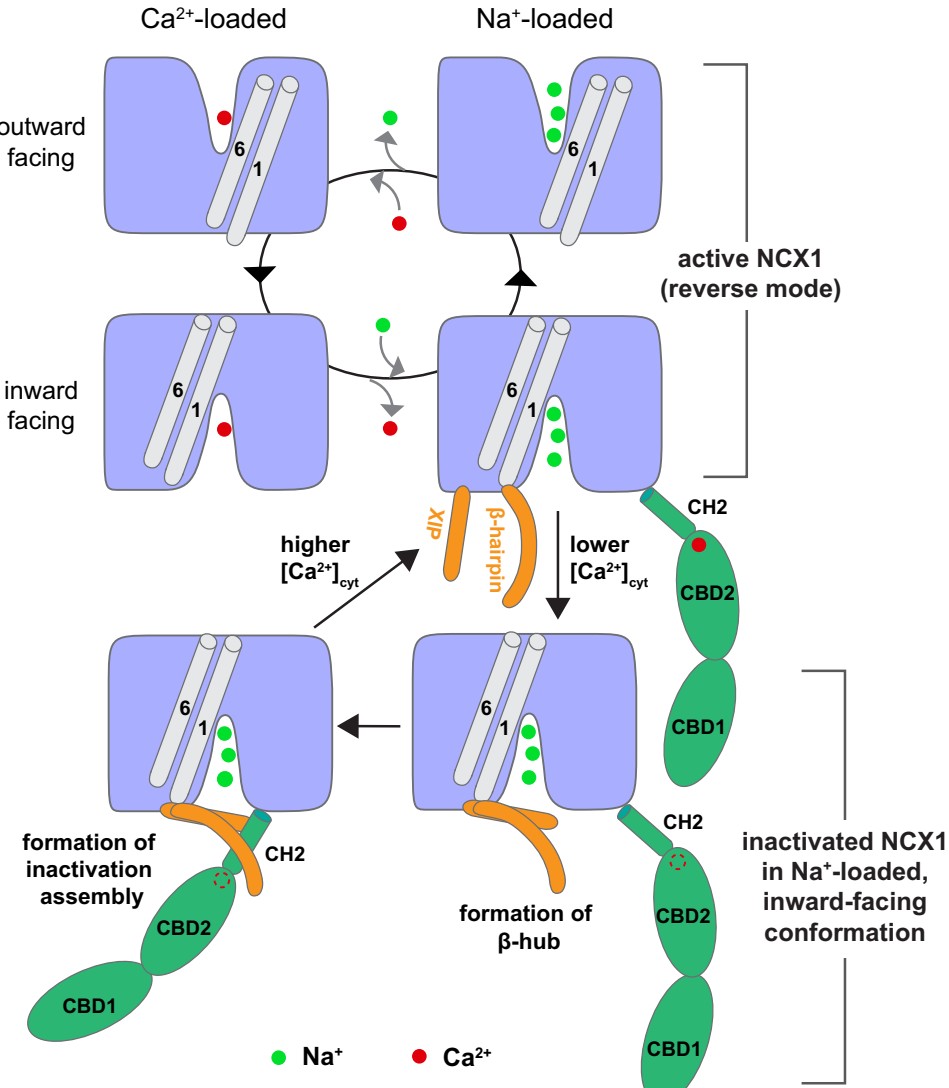

**Fig. 5 | Working model of Na⁺-dependent inactivation and cytosolic Ca²⁺ activation.** The top four states depict the cycle of active NCX1 in reverse mode (Na⁺ efflux). Gray cylinders represent TMs 1 and 6, whose sliding motion controls the inward–outward conformational transition of the exchanger. NCX1 enters the inactivated state by the formation of β-hub between XIP and β-hairpin, which occurs when the exchanger is in the inward-facing, Na⁺-loaded state with low cytosolic [Ca²⁺]. The β-hub subsequently interacts with the cytosolic domain mainly through the CH2 helix of CBD2 and generates a stable inactivation assembly that locks the exchanger in the inward-facing conformation. At higher cytosolic [Ca²⁺], Ca²⁺ binding at CBD2 destabilizes the inactivation assembly and results in the disassembly of the β-hub and the detachment of the cytosolic domain from the TM module. Consequently, the exchanger returns to its active state. For simplicity, XIP (orange rod), β-hairpin (orange sickle), and CBD (green) are only drawn in NCX1 in the Na⁺-loaded, inward-facing state.

fluorescence-based size-exclusion chromatography (FSEC) analysis confirmed the complex formation between NCX1 and 2E4 IgG. To generate the Fab fragment, the purified 2E4 IgG (provided by VGTI) was subject to papain digestion in PBS buffer containing 10 mM EDTA, 10 mM ʟ-Cysteine, 1:100 papain (w/w) for 2 h at 37 °C. The reaction was quenched by adding 30 mM iodoacetamide to the digestion solution and placing it at RT for 30 min in the dark. After verifying the complete papain cleavage by SDS-PAGE, the Fc fragment was separated from Fab using anion exchange chromatography column HiTrap Q, and the flow through containing Fab was collected and concentrated for complex formation and structural analysis.

To obtain the sequence of the 2E4 antibody, total RNA was isolated from the hybridoma using PureLink RNA Mini Kit (Invitrogen) following the manufacturer's protocol. Total RNA was subjected to reverse transcription reactions using Superscript III reverse transcription kit (Invitrogen), and the resulting cDNA was used as a template in PCR reactions with degenerated primers to amplify the antibody variable regions[66]. Sequences of the resulting PCR products with the size of ~500 bp were analyzed using the IMGT database (http://www.imgt.org/) to determine possible variable regions of the light chain and heavy chain. The sequence of the 2E4 clone was verified by mapping the Fv to the EM density of the Fab.

For the generation of NCX1-Fab 2E4 complex, NCX1 in digitonin was incubated with purified Fab in a molar ratio of 1:1.2 (NCX1:2E4 Fab) for 3 h. The samples were further purified by size-exclusion chromatography in buffer A supplemented with 0.06% digitonin. The peak fractions were collected and concentrated to 5–6 mg/ml for cryo-EM analysis. To obtain the Ca²⁺-activated exchanger proteins, 0.5 mM CaCl₂ was added to protein samples 1 h before grid preparation.

**Cryo-EM sample preparation and data acquisition**
Human NCX1-Fab 2E4 samples (5–6 mg/ml) in various conditions were applied to a glow-discharged Quantifoil R1.2/1.3 300-mesh gold holey

carbon grid (Quantifoil, Micro Tools GmbH, Germany), blotted for 4.0 s under 100% humidity at 4 °C and plunged into liquid ethane using a Mark IV Vitrobot (FEI).

For the data of NCX1-Fab (2E4) samples in various states, movies were acquired on a Titan Krios microscope (FEI) operated at 300 kV with a K3 Summit direct electron detector (Gatan), using a slit width of 20 eV on a GIF-Quantum energy filter. Images were recorded with Serial EM in super-resolution counting mode with a super-resolution pixel size of 0.415 Å. The defocus range was set from −0.9 to −2.2 μm. Each movie was recorded for about 5 s in 60 subframes with a total dose of 60 e⁻/Å². All data were collected using the CDS mode of the K3 camera with electron dose rates between 7 and 9 e-/pixel/s.

### Image processing

Cryo-EM data were processed following the general scheme described below with some modifications to different datasets (Supplementary Figs. 2 and 6). First, movie frames were motion-corrected and dose-weighted using MotionCor2[67]. The CTF parameters of the micrographs were estimated using the GCTF program[68]. After CTF estimation, micrographs were manually inspected to remove images with bad defocus values, ice contamination, or carbon. Particles were selected using Gautomatch (Kai Zhang, https://www.mrc-lmb.cam.ac.uk/kzhang/) and extracted using a binning factor of 3. Next, particles were subjected to 2D classification, ab initio modeling, and 3D classification. Classes that showed clear features of the NCX1-Fab complex were subjected to 3D auto-refinement and another round of 3D classification without performing particle alignment using a soft mask around the protein portion of the density.

The best-resolving class was then re-extracted with the original pixel size and further refined. Beam tilt, anisotropic magnification, and per-particle CTF estimations were performed. After the initial processing in Relion 3.1[69,70], the particles were further refined using non-uniform and local refinement in cryoSPARC[71] to improve the resolution of the final reconstruction. The quality of the EM density maps for the TM and cytosolic domains was further improved through focused refinement, allowing for accurate model building for a major part of the protein. All resolution was reported according to the gold-standard Fourier shell correlation (FSC) using the 0.143 criterion[72]. Local resolution was estimated using cryoSPARC.

### Model building, refinement, and validation

The EM map of human NCX1 in the apo inactivated state shows high-quality density for de novo model building in Coot[73], facilitated by the predicted TM structure from Alphafold (based on NCX_Mj) and previous X-ray structure of the CBD domain (PDB: 3US9)[58]. Models were manually adjusted in Coot and refined against maps using the phenix.real_space_refine with secondary structure restraints applied[74]. The final NCX1structural model contains residues 17–248, 370–467, 482–644, 652–698, 707–718, and 738–935. For the bound Fab fragment, only the variable region (Fv) was modeled.

The particles from the NCX1 sample prepared in the presence of 0.5 mM Ca²⁺ were classified into two groups. The apo-inactivated NCX1 structure could be directly docked into the EM map from group I particles without adjustment. The EM maps of NCX1 in the Ca²⁺-activated state from group II particles are relatively poor, particularly at the TM region To model the Ca²⁺-activated structure, the TM structure from apo NCX1 and the X-ray structures of the Ca²⁺-bound CBD1 (PDB: 2DPK) and CBD2 (PDB: 2QVM)[56,57] were fitted into the focused maps of the TM region and cytosolic region, respectively, as rigid-bodies followed by manual adjustment in Coot and refinement in Phenix against the overall map. The statistics of the geometries of the models were generated using MolProbity[75]. All the figures were prepared in PyMol (Schrödinger, LLC), Chimera[76], and ChimeraX[77]. The solvent-accessible cavities were calculated using the program Caver[78].

### Electrophysiological experiments

The wild-type human NCX1 and its mutants were cloned into a pGEMHE vector and expressed in oocytes for electrophysiological recordings. RNA was synthesized using mMessage mMachine (Ambion) and injected into Xenopus laevis oocytes as described in[53]. Oocytes were kept at 18 °C for 4–7 days. Outward human Na⁺–Ca²⁺ exchanger (NCX1) currents were recorded using the giant patch technique in the inside-out configuration. The external solution (pipette solution) contained the following (mM): 100 NMG (N-methylglucamine), 10 HEPES, 20 TEAOH (tetraethyl-ammonium hydroxide), 0.2 niflumic acid, 0.2 ouabain, 8 Ca(OH)₂, pH = 7 (using MES); bath solution (mM): 100 CsOH or 100 NaOH, 20 TEAOH, 10 HEPES, 10 EGTA or HEDTA (N-(2-Hydroxyethyl) ethylenediamine-N,N′,N′-triacetic acid) and different Ca(OH)₂ concentrations to obtain the desired final free Ca²⁺ concentrations, pH = 7 (using MES). Free Ca²⁺ concentrations were calculated according to the MAXc program and confirmed with a Ca²⁺ electrode.

NCX1 currents were evoked by the rapid replacement of 100 mM Cs⁺ with 100 mM Na⁺ using a computer-controlled 20-channel solution switcher. As NCX1 does not transport Cs⁺, there is no current, and only upon application of Na⁺ does the exchange cycle initiate. Data were acquired online at 4 ms/point and filtered at 50 Hz using an 8-pole Bessel filter. Experiments were performed at 35 °C and at a holding potential of 0 mV. The effects of the Na⁺-dependent inactivation were analyzed by measuring fractional currents calculated as the ratio of the steady-state current to the peak current (fractional activity) for the recordings at 100 mM Na⁺ and 12 μM Ca²⁺. All P values were calculated using an unpaired, two-sided Welch's t-test.

### Reporting summary

Further information on research design is available in the Nature Portfolio Reporting Summary linked to this article.

### Data availability

The cryo-EM density maps of the human NCX1 have been deposited in the Electron Microscopy Data Bank under accession numbers EMD-40457 for the apo inactivated state and EMD-40460 and EMD-40467 for the group I and II structures in the presence of 0.5 mM Ca²⁺, respectively. Atomic coordinates have been deposited in the Protein Data Bank under accession numbers 8SGJ for the apo inactivated structure and 8SGT for the group II Ca²⁺-bound activated structure. The source data underlying Figs. 1c and 3c are provided with this paper as a Source data file. Source data are provided in this paper.

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

## Acknowledgements

Single particle cryo-EM data were collected at the University of Texas Southwestern Medical Center Cryo-EM Facility, which is funded by the CPRIT Core Facility Support Award RP170644. Cryo-EM sample grids were prepared at the Structural Biology Laboratory at UT Southwestern Medical Center, which is partially supported by grant RP170644 from CPRIT. This work was supported in part by the Howard Hughes Medical Institute (to Y.J.) and by grants from the National Institute of Health (R35GM140892 to Y.J. and R01HL152296 to M.O.) and the Welch Foundation (Grant I-1578 to Y.J.).

## Author contributions

J.X. prepared the samples; J.X. and Y.H. performed data acquisition, image processing, and structure determination; M.O., S.J. and W.Z. performed electrophysiology recording; Y.J. supervised the work; all authors participated in research design, data analysis, discussion, and paper preparation.

## Competing interests

The authors declare no competing interests
