## [Peer Review File · Nature Communications]

Structural mechanisms of the human cardiac sodium-calcium exchanger NCX1Reviewers' Comments:

Reviewer #1:

Remarks to the Author:

This paper presents the first published structures of a eukaryotic Na-Ca exchanger (NCX) determined by cryoEM. The investigators chose human NCX1 as this a functionally well-documented Na-Ca exchanger critical for proper cardiac function in humans. The highest resolution was obtained for the inward facing NCX1 in the so-called sodium-inactivated state. A lower resolution structure of the calcium-activated outward facing NCX1 is presented as well. The only published NCX structure to date is that of the crystal structure of the distantly related archaebacterial exchanger NCX_Mj, obtained in the lab of the senior author of the present study Dr. Jiang. The high-resolution structure of the inward-facing and sodium-inactivated cardiac exchanger provides important new information about the overall structure of the exchanger, in particular the interaction between the transmembrane domain and the large hydrophilic loop located in the cytosol which is important in regulation of Na-Ca exchange transport via sodium-dependent inactivation as well as activation by cytosolic free calcium concentration.

I should preface my comments by stating that I am not able to evaluate some of the technical work that went into obtaining the NCX1 structures presented here. Nevertheless, I believe this is a most important study that provides the first high-resolution structure of a eukaryotic Na-Ca exchanger and allows for many new insights in the operation of this functionally critical protein in the human heart. The paper is well-written with an extensive and appropriate list of references, and I believe is most suitable for publication in Nature Communications.

A few comments and questions:

1. To assist in obtaining structures the authors introduce some small domain mutations and verify that functional activity is retained or modified as desired (Figs. 1c and 3c) using electrophysiological methods. Have the authors any information on whether the purified NCX1 proteins used for the cryoEM studies retained functional activity?
2. It would be helpful to have more extensive figure legends so one can understand each figure as a standalone item.
3. The TMH1 and TMH2 domains that precede TM6 were initially based on hydropathy analysis considered to be part of a transmembrane helix and it is interesting to note that in the new structure they are embedded in the membrane. I believe they are found in most eukaryotic NCX sequences. The authors suggest that "they are an integral part of the TM domain and likely move together with TM1 and TM6 during inward -outward conformational changes". However, it is noteworthy that the NCX_Mj sequence (and other prokaryotic NCX sequences) do not contain these segments and I wonder if the authors have any thoughts about the significance of this? Could this be part of the reason that NCX proteins have significantly greater turnover numbers of $\sim 5000/S$ compared with NCX_Mj (0.5/S).
4. Found one typo. "Albert" (page 10, line 3) should probably be "Albeit".
5. I was less convinced about the section concerning the structural mechanism of cytosolic calcium activation based on structures obtained in the presence of high calcium. How was this sample prepared compared to samples that are thought to represent the sodium-inactivated state?

Reviewer #2:

Remarks to the Author:

Xue et al. present a study that sheds light on the structures of human NCX1 in both the activated and inactivated states. Human NCX1 plays a crucial role in human health, and this study contributes significantly to our mechanistic understanding. Using single-particle electron cryomicroscopy (cryo-EM), the authors provide the first detailed structure of human NCX1.

The execution of this study is at a high level, as the authors demonstrate distinct interactions between the transmembrane domain (TMD) and cytosolic domain in the inactivated state compared to the active state. The authors employed point-mutagenesis and deletion mutations in conjunction with

giant patch-clamp electrophysiology to validate this observation. This approach reinforces the findings and highlights the tight interaction between the TMD and cytosolic domain in the absence of calcium, which gives rise to the inactivation assembly. This assembly appears to 'lock' the conformation and limit the motion of the TMD transporter during inactivation.

The proposed functional mechanism is fascinating and well-supported by the functional assay used in the study. In conclusion, Xue et al.'s study offers a remarkable contribution and should be published with minor revisions.

Overall, the manuscript's primary point is solid, but it requires some improvements, particularly in addressing the following issues:

1. **Statistical analyses:** The manuscript lacks statistical analyses for electrophysiology data in Fig. 1c and Fig. 3c. The authors should conduct pairwise comparisons to assess their findings and ensure the results are statistically robust.
2. **Recording conditions:** It is unclear whether all recordings were done at 0 mV, as described in the methods section. To clarify this, the authors could add this information in the figure legend for Fig. 1c and Fig. 3c.
3. **Potential functional effects of Fab:** The manuscript could address the potential functional impact of Fab's presence in the sensitive area of the CBD on the electrophysiological functions studied. However, I am aware that this electrophysiology may be technically tricky to conduct, so if not possible, the experiment will not have to be included in this manuscript.
4. **Missing legends:** There are some missing legends, particularly for Fig. 5 and Extended Fig. 1. If the absence of legends is intentional, the authors should add explanatory sentences to provide context and make the figures more understandable.

To Reviewer #1:

We appreciate reviewer #1's positive comments and the points raised by the reviewer are addressed as follows:

1. To assist in obtaining structures the authors introduce some small domain mutations and verify that functional activity is retained or modified as desired (Figs. 1c and 3c) using electrophysiological methods. Have the authors any information on whether the purified NCX1 proteins used for the cryoEM studies retained functional activity?

We used the electrophysiological recording of the EM construct expressed in *Xenopus oocytes* to demonstrate that it has similar functionality as the wild-type exchanger (Fig. 1c). We feel the use of electrophysiology provides a more quantitative measurement of eukaryotic exchanger activity. We have not tried to reconstitute the purified protein in liposomes for functional assay but expect the mutant would retain the ion-exchanging activity in proteoliposomes. However, the functional assay using proteoliposomes may not be feasible to measure Ca^{2+} activation and Na^{+} -dependent inactivation of the exchanger.

2. It would be helpful to have more extensive figure legends so one can understand each figure as a standalone item.

We have included more detailed descriptions of the figures in the figure legends in the revision, primarily for Fig. 5 and Supplementary Fig. 1. In addition, we also added four videos in the revision to help readers visualize the structures and conformational changes described in the manuscript.

3. The TMH1 and TMH2 domains that precede TM6 were initially based on hydropathy analysis considered to be part of a transmembrane helix and it is interesting to note that in the new structure they are embedded in the membrane. I believe they are found in most eukaryotic NCX sequences. The authors suggest that “they are an integral part of the TM domain and likely move together with TM1 and TM6 during inward-outward conformational changes”. However, it is noteworthy that the NCX_Mj sequence (and other prokaryotic NCX sequences) do not contain these segments and I wonder if the authors have any thoughts about the significance of this? Could this be part of the reason that NCX proteins have significantly greater turnover numbers of ~5000/S compared with NCX_Mj (0.5/S).

The TMH1 and TMH2 regions may influence the turnover rate of the mammalian NCX since they house the palmitoylation site at C738 which affects the Na^{+} -dependent inactivation. However, there are limited studies on mutations within TMH1 and TMH2 regions to implicate their involvement in NCX turnover rate. Fuller's group has mutated residues 738 and 739 (distal TMH2) and deleted parts of these regions. They only performed palmitoylation and trafficking studies without electrophysiological data (Plain F, et al., An amphipathic α -helix directs palmitoylation of the large intracellular loop of the sodium/calcium exchanger. *J Biol Chem.* 2017 292:10745-10752.). Philipson's group mutated residue C738 to alanine and the resulting

exchanger had the same activity as WT when investigated using Na⁺-dependent ⁴⁵Ca²⁺ uptake. His group also inserted an alanine at either position 701 or 749 to disrupt the alpha helices and the two resulting exchangers were active (Nicoll D.A., et al ., A New Topological Model of the Cardiac Sarcolemmal Na⁺-Ca²⁺ Exchanger J Biol Chem, 1999, 274: 910-917 DOI: <https://doi.org/10.1074/jbc.274.2.910>). Since mammalian NCX and NCX Mj are different in many aspects, we do not know if the lack of TMH1/2 in NCX Mj partly contributes to the low turnover rate of the archaea exchanger. It is worth noting that the thermophilic archaea have different membrane composition than mammals and NCX Mj from *Methanococcus jannaschii* may have a higher turnover rate when measured at its high native temperature.

4. Found one typo. “Albert” (page 10, line 3) should probably be “Albeit”.

Thanks for pointing out. It has been corrected in the revision.

5. I was less convinced about the section concerning the structural mechanism of cytosolic calcium activation based on structures obtained in the presence of high calcium. How was this sample prepared compared to samples that are thought to represent the sodium-inactivated state?

The protein preparations are the same for all samples used in the structural determination of this study. The only difference between the samples for the inactivated state and the Ca²⁺ activated state is that no Ca²⁺ was added in the former sample whereas 0.5 mM of Ca²⁺ was added in the latter protein sample just before preparing the cryo grids for EM data collection. Thus, structural changes between the two states have to be caused by the Ca²⁺. Consistently, we only observe Ca²⁺ binding in CBD2 with the addition of Ca²⁺. Although at a lower resolution, the single particle analysis clearly indicates that the Ca²⁺-bound NCX1 is highly dynamic and inactivation assembly is destabilized.

To Reviewer #2:

We appreciate reviewer #2's positive comments and constructive suggestions. The points raised by the reviewer are addressed as follows:

1. Statistical analyses: The manuscript lacks statistical analyses for electrophysiology data in Fig. 1c and Fig. 3c. The authors should conduct pairwise comparisons to assess their findings and ensure the results are statistically robust.

As suggested, the number of measurements (n) and P-values for comparison between any two given groups were included in Fig. 1c and Fig. 3c.

2. Recording conditions: It is unclear whether all recordings were done at 0 mV, as described in the methods section. To clarify this, the authors could add this information in the figure legend for Fig. 1c and Fig. 3c.

All recordings were performed at 35 °C with a holding potential of 0 mV. This information has been added in the legends for Fig. 1c and Fig. 3c as suggested.

3. Potential functional effects of Fab: The manuscript could address the potential functional impact of Fab's presence in the sensitive area of the CBD on the electrophysiological functions studied. However, I am aware that this electrophysiology may be technically tricky to conduct, so if not possible, the experiment will not have to be included in this manuscript.

We have not investigated the functional impact of Fab on NCX1 currents due to technical concerns, as the reviewer recognized. Our giant patch recordings are performed at 35 °C by continuously perfusing the tip of the patch pipette with recording solutions. These heated solutions are delivered with a custom-made perfusion system with a flow rate of about 5 mL/min. Every single trace shown in the manuscript requires a minimum of ~ 20/30 mL of recording solution. As such, testing the Fab effects would require such large quantities of Fab that are unattainable, especially in consideration that its effects will have to be investigated in multiple conditions such as in the presence of different Ca²⁺ concentrations. We simply don't have sufficient antibody material to perform this study.

4. Missing legends: There are some missing legends, particularly for Fig. 5 and Extended Fig. 1. If the absence of legends is intentional, the authors should add explanatory sentences to provide context and make the figures more understandable.

In the revision, we have included more detailed descriptions of figures (primarily for Fig. 5 and Supplementary Fig. 1) in the figure legends. In addition, we also added four videos in the revision to help readers visualize the structures and conformational changes described in the manuscript.

Reviewers' Comments:

Reviewer #1:

Remarks to the Author:

The authors have addressed the reviewer's comments in a satisfactory way, and I have no further comments or concerns. This is a very important contribution in our understanding of the structure and function of the cardiac Na/Ca exchanger and worthy to be published in Nature Communications. I found the addition of the video's to be very helpful.

Reviewer #2:

Remarks to the Author:

The authors addressed concerns raised by me and reviewer1. This manuscript is ready to be published.

REVIEWERS' COMMENTS

Reviewer #1 (Remarks to the Author):

The authors have addressed the reviewer's comments in a satisfactory way, and I have no further comments or concerns. This is a very important contribution in our understanding of the structure and function of the cardiac Na/Ca exchanger and worthy to be published in Nature Communications. I found the addition of the video's to be very helpful.

We appreciate the reviewer's enthusiastic and supportive comments.

Reviewer #2 (Remarks to the Author):

The authors addressed concerns raised by me and reviewer1. This manuscript is ready to be published.

We appreciate the support from this reviewer.